# Preparation and Characterization of Soy Protein Isolate-Based Nanocomposite Films with Cellulose Nanofibers and Nano-Silica via Silane Grafting

**DOI:** 10.3390/polym11111835

**Published:** 2019-11-07

**Authors:** Zhiyong Qin, Liuting Mo, Murong Liao, Hua He, Jianping Sun

**Affiliations:** School of Resources, Environment and Materials, Guangxi university, Nanning 530000, China; q_zhiyong@163.com (Z.Q.); mo_liuting@163.com (L.M.); mr_liaomr@163.com (M.L.)

**Keywords:** soy protein, nano-silica, interface design, synergistic enhancement, mechanical properties, water resistance

## Abstract

Soy protein isolate (SPI) has attracted considerable attention in the field of packaging technology due to its easy processability, biodegradability, and good film-forming characteristics. However, SPI-based films often suffer from inferior mechanical properties and high moisture sensitivity, thus restricting their practical application. In the present study, herein, a biobased nanocomposite film was developed by cross-linking SPI matrix from the synergistic reinforcement of cellulose nanofibers (CNF) and nano-silica (NS) particles. First, we functionalized the CNF with NS using a silane agent (KH560) as an efficient platform to enhance the interfacial interaction between SPI and CNF/NS, resulting from the epoxy-dominated cross-linking reaction. The chemical structure, thermal stability, and morphology of the resultant nanocomposite films were comprehensively investigated via Fourier transform infrared (FTIR) spectroscopy, X-ray diffraction (XRD), scanning electron microscopy (SEM), and thermogravimetric analysis (TGA). These results supported successful surface modification and indicated that the surface-tailored CNF/NS nanohybrid possesses excellent adhesion with SPI matrix through covalent and hydrogen-bonding interactions. The integration of CNF/NS into SPI resulted in nanocomposite films with an improved tensile strength (6.65 MPa), representing a 90.54% increase compared with the pristine SPI film. Moreover, the resulting composites had a significantly decreased water vapor permeation and a higher water contact angle (91.75°) than that of the unmodified film. The proposed strategy of synergistic reinforcements in the biobased composites may be a promising and green approach to address the critical limitations of plant protein-based materials in practical applications.

## 1. Introduction

Under the increasing environmental pressure due to plastic pollution and the depletion of oil resources, the development of environmentally friendly and biodegradable materials from low-cost or freely available natural and renewable resources, such as agricultural and food waste products, has received considerable attention [1]. Proteins, one of the most important natural macromolecules, including collagen, milk protein, egg protein, fish protein, silk fibroin, keratin from animals as well as soy protein isolate (SPI), corn gliadin, and wheat protein from plants, provide a wide range of potential functional properties as a biodegradable membrane through the formation of numerous intermolecular bonds due to the presence of polar functional groups [2,3,4]. SPI, as an edible, available, wearable, and degradable plant protein resource, not only plays an important role in the traditional industry of food, nutrition and health care products, but also shows broader application prospects in the field of emerging materials, including non-formaldehyde wood adhesive, biodegradable composites, and food packing material applications [5,6,7]. However, the poor mechanical properties and water resistance of soy protein materials limit their successful applications in many fields. Many efforts have been made to improve the inherent properties of soy-based materials, including chemical/physical cross-linking, surface modification, and biomimetic structure design. The mechanical properties and water resistance of the prepared films have been improved, but it is still far from what we want. In addition, SPI also cannot resist high temperature or strong acid/base with different materials.

Currently, nanohybrid reinforcement is among the rapidly emerging techniques being implemented in the modification of biopolymers, as well as improving their physicochemical/mechanical properties. Common organic/inorganic nanofillers include carbon nanoparticle [8,9,10], cellulose nanocrystal [11], carboxymethyl cellulose [12], starch nanocrystal [13], montmorillonite [14,15,16], silver nanoparticle [17], TiO_2_ [18], and so on. They would enhance the mechanical properties and thermal stability of composite materials via the formation of hydrogen bonds and cross-linking system. Nano-silica (NS) has been regarded as a promising reinforcement filler for polymer composites because of its higher specific surface area, good mechanical properties, and low cost. However, the inherent high surface energy and nano assembly tendency of NS result in low dispersibility and weak properties of NS-matrix interface, and thus the lack of compatibility with polymer matrix that impairs the enhancement of mechanical and barrier properties [19,20,21].

To high-efficiently utilize the potential of NS as reinforcement in composites, rational surface structure designs are essential for altering the interfacial performance of NS matrix. In this respect, cellulose nanofibers (CNF) have shown promise as green dispersing agents for various nanoparticles such as graphene oxide, carbon nanotubes [22], silver nanoparticles [23], and PMMA–*b*–PAA complex [24]. Different from other natural nanocrystals (e.g., CNCs or SNCs), CNF hold a special net-like morphology with typical interspersed crystalline regions throughout the nanofibrils, thus endowing them higher specific surface area, excellent mechanical properties, and superior efficient dispersing effect. In addition, the prevalence of hydroxy groups on the surface of CNF provide active sites for further functionalization. For instance, the deposition of nanoparticles could introduce many functional motifs onto the surface of nanofibrils, resulting in remarkable improvement of fiber-matrix interphase properties. Therefore, considering the inherent high strength and interesting dispersing capacity of CNF, designing the synergistic reinforcement of such CNF/NS nanohybrids for high-performance biobased composites is very attractive.

Herein, we introduce CNF/NS into SPI nanocomposite films to promote the homogeneous dispersion of NS and potentially improve the mechanical properties, water resistance, and barrier properties of the SPI-based composites. First, the surface-tailored CNF/NS (O@CNF/NS) nanohybrid was prepared via one-step co-deposition of γ-(2,3-epoxypropoxy)propytrimethoxysilane (KH560) on the CNF/NS surface (Scheme 1). The KH 560, known as versatile coupling agents, could firmly introduce the epoxy groups onto the nanohybrid surface. Next, the solution casting method was employed to prepare SPI/O@CNF/NS nanocomposite films. The effect of the incorporation of O@CNF/NS on the surface compositions, microstructure, and thermal properties of the films was investigated, and the reaction mechanism between O@CNF/NS and SPI is proposed. The water resistance, mechanical and barrier properties of the resultant films were thoroughly analyzed.

## 2. Materials and Methods

### 2.1. Materials

SPI with protein content over 95% was obtained from Yuwang Ecological Food Industry Co., Ltd. (Shandong, China). γ-(2,3-epoxypropoxy)propytrimethoxysilane (KH560) (purity ≥ 99%) and Glycerol were purchased from McLean Biochemical Reagent Co., Ltd. (Shanghai, China). CNF with a diameter of 4–10 nm and length of 1000–3000 nm was provided by Qihong Technology Co., Ltd. (Guilin, China). Nano-silica (purity ≥ 99.8%) with a particle size of 15 ± 5 nm was acquired from Maikun Chemical Co., Ltd. (Shanghai, China). Sodium hydroxide (purity ≥ 97%) was purchased from Kelon Chemical Reagent Factory (Chengdu, China). All reagents were of analytical grade and used as received.

### 2.2. Surface Modification of the O@CNF/NS by Silane Grafting

First, 2.0 g of KH560 was hydrolyzed in 100 mL of mixture (ethanol: water, 85:15, *v*/*v*), which was adjusted to pH (4.0) with acetic acid (20%, *w*/*w*). CNF (0.15 g) and NS (0.15 g) powders were added to the suspension and magnetically stirred at 80 °C for 8 h. After centrifugation, the resultant O@CNF/NS hybrids were washed three times with ethanol/distilled water (1:1, *v*/*v*) and dried at 90 °C to a constant weight.

### 2.3. Preparation of SPI/O@CNF/NS Films

The SPI-based nanocomposite films were prepared using a solution-cast method. Desired amounts of CNF/NS were mixed with SPI in deionized water by magnetically stirring for 10 min, followed by ultrasonic dispersion in an ice bath for 20 min to achieve homogeneous dispersion. Subsequently, glycerol (30 wt % of total SPI weight) was added as plasticizer, the pH of the suspensions was adjusted to 9.0 ± 0.2 using 10 wt % NaOH solution to denature the SPI molecules and immediately heat-treated at 80 °C for 30 min. After heating, the resulting solution was poured into plastic petri dishes and dried at 45 °C and 45 RH% for 24 h to prepare composite films. All of the films were conditioned in a desiccator (57 ± 2 RH% and 25 ± 2 °C) days before characterization. For comparison, pristine SPI film and NS, CNF, and CNF-NS modified SPI films were also prepared using the same procedure. Detailed code and formulations of all the composite films are described in Table 1.

### 2.4. Characterization of SPI/O@CNF/NS Composite Films

#### 2.4.1. Mechanical Properties

The dimension of each film was measured using a handheld digital micrometer (Measuring & Cutting Tool Co., Ltd. Guilin, China) with a sensitivity of 0.001 mm. The mechanical properties of the films (10 mm × 60 mm) were investigated according to the ASTM D880-02 on a universal tensile testing system (AG-X Plus, Shimadzu Corporation, Kyoto, Japan) with a tensile rate of 50 mm/min and gauge length of 30 mm. The tensile strength, strain at break, and Young’s modulus were determined from the testing results. Each sample was tested six times [10].

#### 2.4.2. Total Soluble Matter

The total soluble matter of each film was measured according to a method described by Xu et al [25]. All specimens with dimensions of 20 mm × 20 mm square-shape were tested with the method. First, all specimens were dried in an air-circulating oven at 103 ± 2 °C until they reached a constant weight, weighing $\left(m_{1}\right)$. Then, the specimens were placed in distillation bottles containing distilled water respectively, with magnetic stirring every 2 h for 24 h at room temperature (25 ± 2 °C). Finally, the specimens were dried in an air-circulating oven at 103 ± 2 °C again, until they reached a constant weight, weighing $\left(m_{2}\right)$. The total soluble matter of each film was calculated as follows:(1)$$TSM\left(\%\right)=\frac{m_{1}-m_{2}}{m_{1}}\times 100$$

#### 2.4.3. Water Uptake

Water resistance property of the films was evaluated through water uptake (WU). The film specimens (20 mm × 20 mm) were dried in an air-circulating oven at 103 ± 2 °C to a constant weight. The samples were placed in a desiccator (25 ± 2 °C, 92 RH%) for a period of 48 h. The WU was calculated by the following equation as follows:(2)$$WU\left(\%\right)=\frac{m_{t}-m_{3}}{m_{3}}\times 100$$

Since the thickness of each film was much smaller than its width, the diffusion of water molecules in the film can be regarded as one-dimensional diffusion [26]. So, the average water absorption could be calculated by Equation (3):(3)$$\frac{M_{t}}{M_{\infty }}=1-\sum_{n=0}^{\infty }\frac{8}{{\left(2n+1\right)}^{2}\pi^{2}}\exp\frac{\left(-{\left(2n+1\right)}^{2}\pi^{2}Dt\right)}{l^{2}}$$
where $M_{\infty }$ is the balance of weight (g), $M_{t}$ is the weight of different time (g), l is the film thickness (mm), D is the diffusion coefficient (cm^2^/s), and t represents the time. If $\frac{M_{t}}{M_{\infty }}$ ≤ 0.5, the Equation (3) could be changed to Equation (4), with a deviation of 0.1%.

(4)MtM∞=4l(Dπ)12t12

Then, taking $\frac{M_{t}}{M_{\infty }}$ = 0.5 to calculate the diffusion coefficient, Equation (5) was obtained.

(5)$$D=\frac{\pi l^{2}}{64t}$$

#### 2.4.4. Water Vapor Permeability (WVP)

The water vapor permeability (WVP) of specimens was examined according to previously reported research [27]. The samples with a diameter of 5 cm were horizontally mounted over the test cup containing anhydrous calcium chloride (0 RH%). The tested cup was then conditioned in a desiccator with saturated copper sulfate solution (97 RH%). At specific time intervals of 2 h, the change of the cup weight was recorded over 2 days according to the following formula:(6)$$WVP=\frac{\Delta M\times d}{A\times t\times \Delta P}$$
where ΔM is the increased weight of the bottle (g), d is film thickness (m), A is the exposed area (m^2^), t is the time lag for permeation (h), and ΔP is water vapor partial pressure difference across the film (Pa). All experiments were carried out in triplicate.

#### 2.4.5. Surface Contact Angle

The contact angle test films were conducted on a DSA100 contact angle meter (KRUSS Co., Ltd., Hamburg, Germany) in an environment of 57 ± 2 RH% at 25 ± 2 °C. The film with a size of 20 mm × 80 mm was fixed on a slide horizontally, and then 5-µL water droplets were dropped on the surface of the specimen using a digital microsyringe. The dynamic process of the water droplet on the specimen surface was recorded. Six parallel measurements were carried out for each sample.

#### 2.4.6. Fourier Transform Infrared Spectroscopy Analysis (FTIR)

Fourier transform infrared spectroscopy (FTIR) was recorded using a FTIR spectroscopy (Nicolet 7600 Nico-let Instrument Corporation, Madison, WI, USA) over the 650–4000 cm^−1^ region at 32 scans and a resolution of 4 cm^−1^.

#### 2.4.7. X-Ray Diffraction (XRD)

X-ray diffraction (XRD) analysis was conducted using an XRD instrument (MINFLEX600, science company Co., Ltd., Shanghai, China) equipped with Cu Kα radiation (λ = 0.154 nm) at 40 kV and 40 mA. The data were collected from 5° to 60° at a scan rate of 2 °/min and a step interval of 0.02°.

#### 2.4.8. Thermogravimetric Analysis (TGA)

The thermal properties of the SPI-based films were investigated using thermogravimetric analysis (TGA, Q50, TA Instruments, MA, USA). The specimens were conditioned in desiccators (0% relative humidity) with P_2_O_5_ desiccant at room temperature for a constant weight prior to the test. The specimens were then heated from room temperature to 600 °C at a heating rate of 10 °C/min under a flow of 60 mL/min nitrogen gas. The peak temperature at maximum degradation rate for each sample was calculated.

#### 2.4.9. Scanning Electron Microscopy (SEM) and Energy Dispersive X-ray Spectroscopy (EDX)

SEM and EDX analysis (ZEISS, Industrial Measurement Technology Co., Ltd., Shanghai, China) were employed to characterize the cross-sectional fracture surface morphologies of the SPI-based films. The cross-sectional fracture was first sputter-coated with a gold layer and then observed using SEM at an acceleration voltage of 15 kV.

## 3. Results

### 3.1. Structural Analysis

The effect of the O@CNF/NS on the protein structure and functional groups was first investigated using FTIR spectroscopy. As shown in Figure 1, the FTIR spectra of the pristine SPI exhibited typical characteristic absorption peaks at 1651, 1544, and 1240 cm^−1^ wavenumbers, assigned to amide I (C=O stretching), amide II (N–H bending), and amide III (C–N and N–H stretching), respectively [10,27]. Additional peaks were observed at 2937, 3281, and 1032 cm^−1^, ascribed to the stretching vibrations of –CH_2_, –OH vibrations, and C–O stretching of glycerol, respectively [28]. Compared to the pristine SPI, the SPI/O@CNF/NS exhibits similar spectrum with several new peaks appearing at 1032 and 914 cm^−1^ (Si–O–Si stretching vibrations), indicating the desired compatibility between the incorporated O@CNF/NS and protein matrix [29]. Furthermore, it is found that the characteristic absorption peak of the epoxy group at 910 cm^−1^ disappears, and the peak at 1240 cm^−1^ attributed to C–N stretching vibration shows an obvious shift in the SPI/O@CNF/NS. These results suggest that the O@CNF/NS forms the epoxy ring-opening polymerization reaction with protein as well as induces additional physicochemical cross-linking interactions within SPI/O@CNF/NS.

The XRD measurement was employed to further investigate the effect of O@CNF/NS on the protein structure as shown in Figure 2. The pristine SPI presents two characteristic crystallization diffraction peaks at 8.8° and 19.8° corresponded to α-helix and β-sheet structures of the SPI secondary structure, respectively, which are in agreement with previous results [27,29]. With the incorporation of CNF/NS, the intensity of the diffraction peaks at 8.8°decreases due to the formation of hydrogen bonding between CNF/NS and SPI increases the content of irregular crimp structure thus leading to a loose and disordered SPI structure. In comparison with the CNF/NS, it is concluded that the diffraction peaks at 8.8° disappeared and the diffraction peak at 19.8° became wider and smoother for the SPI/O@CNF/NS [11]. This could be explained that the introduction of KH560 further accelerate the physical/chemical interactions between CNF/NS and SPI, which significantly increases the overall crosslinking degree thus unbalancing the ordered array of SPI molecule for a deceased crystalline degree as supported by the results of FTIR analysis.

### 3.2. Thermal Stabilities Analysis

The effect of the O@CNF/NS on the thermal properties of SPI films was studied using the TG measurement, of which the results are shown in Figure 3 and Table 2. It is found that the pristine SPI presents three degradation processes in the range of 30–130, 130–270, and 270–500 °C, which are attributed to the evaporation of adsorbed water, the evaporation of glycerol, and the thermal degradation of proteins chains, respectively [15]. Compared to the pristine SPI, the addition of CNF, NS, or CNF/NS all obviously increases the residual weight and decreases the maximum degradation rate. This could be explained due to the fact that the incorporated nanofiller not only forms interactions with protein for an improved crosslinking system but also accelerates the energy transfer resulting in thermal-stability composites. Most importantly, it is obviously observed that the introduction of O@CNF/NS further decreases the degradation rate, reaching the lowest degradation rate in SPI/O@CNF/NS in comparison with other samples. This proves the improvement of the thermal stability of SPI/O@CNF/NS, which is mainly because the epoxy groups induce further interactions between nanofillers and protein thus constructing a stable crosslinking network to resist thermodestruction [30].

### 3.3. Morphology

The surface morphology of SPI/O@CNF/NS was examined using SEM and EDX tests to study the dispersibility and interfacial interactions of the nanoparticles in SPI matrix. As presented in Figure 4, the pristine SPI exhibits relatively coarse fracture surfaces with numerous holes in accordance with previous reports [10,31], and the silicon content is very small. It is observed that the incorporation of CNF or NS results in a dense and smooth cross-sectional fracture in SPI/CNF or SPI/NS due to the CNF or NS filling the discontinuous matrix and forming interactions with protein for a dense system, while the CNF aggregation in SPI/CNF is owing to the poor interfacial activity of CNF [31]. The integration of CNF and NS shows a desired synergistic effect in SPI, which effectively improves the dispersion of CNF and forms an even and uniform fracture in SPI/CNF/NS. Moreover, compared with the others, the SPI/O@CNF/NS film exhibits a denser morphology with several light and spotted points, which reveals that the KH560 improves the interactivity of O@CNF/NS to form multiple interactions with SPI matrix thus obtaining an optimized dispersion in SPI; the uniform distribution of silicon elements in the EDX image also proves this view.

### 3.4. Mechanical Properties

As an important factor for potential applications, the overall mechanical properties of SPI has been effectively improved via incorporating nanofillers or constructing chemical crosslinking network [32,33]. The effect of incorporating CNF, NS, CNF/NS, and O@CNF/NS in the mechanical properties of SPI was studied in the tensile strength (TS), elongation at break (EB), and Young’s modulus tests. As concluded from Table 3 and Figure 5, the introduction of NS results in a 20.92% increment in TS of the SPI/NS compared to the pristine SPI due to the strong hydrogen bonding formation between SPI and NS nanoparticles (Scheme 2). It is found that, in SPI/CNF, the introduction of CNF further exhibits a 41.55% increment of TS value in comparison with SPI/NS owing to the formation of the strong hydrogen bonding and the desired physical cross-linking network between SPI and CNF. However, the EB obviously decreases from 153.54% to 117.45% because its poor surface activity leads to an inferior interfacial stress transfer. For the SPI/CNF/NS, the benefit of the integration of CNF and NS is the improved ability of stress transfer between nanofillers and protein interface thus synergistically optimizing the mechanical property of the material. As a result, the TS of the pristine SPI is increased from 3.49 MPa to 4.44 MPa, while the EB value shows a slight decrease in SPI/CNF/NS. Moreover, the TS value reaches the maximum of 6.65 MPa after adding the O@CNF/NS, realizing a 90.54% increment compared to the pristine SPI, while the EB value exhibits a decrement from 153.54% to 109.62%. This improvement is mainly because the KH560 not only bridges the CNF and NS for an excellent stress-transfer skeleton but also induces CNF/NS to form chemical interactions with protein for a stable crosslinking network with the assistance of epoxy ring-open crosslinking reactions. The increment of crosslinking degree within SPI/O@CNF/NS can lead to a rigid composite system and thus the decrement of EB compared to the pristine as supported by the FTIR and XRD results, of which the enhancement mechanism within SPI/O@CNF/NS is illustrated in Scheme 2.

### 3.5. Water Resistance Properties of SPI-based Films

The moisture content (MC), total soluble matter (TSM), water uptake (WU), diffusion coefficient (DC), and water vapor permeability (WVP) were measured to investigate the effect of the O@CNF/NS on SPI, of which the results are shown in Table 4. It is found that the MCs of the pristine SPI exhibits a decrement with the addition of CNF, CNF/NS, and O@CNF/NS, which is mainly due to the polyhydroxy CNF. The TSM test is related to the molecular structure of the SPI-based composites, which can reflect the physical and chemical interactions between SPI and nanofillers. In comparison with the pristine SPI, the TSM value of the SPI/NS, SPI/CNF, SPI/CNF/NS, and SPI/O@CNF/NS present a decrement of 21.84%, 22.96%, 22.61%, and 27.05%, respectively, mainly due to the formation of multiple physical and chemical interactions between SPI matrix and nanofillers for a stable crosslinking structure. In general, the WU process of SPI-based composites can be divided into two stages, containing the surface adsorption and the internal diffusion permeability. As shown in Figure 6, the introduction of NS, CNF, and CNF/NS leads to a decrement in WU of 5.7%, 11.9%, and 10.6%, respectively, reaching the maximum of 19.5% in SPI/O@CNF/NS film compared to the pristine SPI. It can be explained by the fact that the incorporated nanofillers forms multiple physical and chemical reactions thus decreasing the gap within protein molecular chains to hinder water permeation. The diffusion coefficient is calculated using Equation (6) and the diffusion coefficient of the pristine SPI film is 2.00 × 10^−10^ cm^2^/s. Compared to the pristine SPI, the DC value exhibits a decrement with the introduction of CNF, CNF/NS, or O@CNF/NS due to the formed hydrogen bonding and chemical crosslinking between nanofillers and protein, making it more difficult for water to diffuse into the film. The WVP was also employed to further investigate the effect of O@CNF/NS on the water resistance of SPI composites. In comparison with other samples, the addition of O@CNF/NS significantly decreases the WVP of pristine SPI from 5.6 × 10^−15^g·cm^−1^·s^−1^·MPa^−1^ to the minimum of 5.1 × 10^−15^g·cm^−1^·s^−1^·MPa^−1^ mainly because KH560 endows the CNC/NS with improved interfacial interactivity thus further synergistically strengthening SPI for a stable cross-linking network structure. These results prove that the water resistance of SPI is improved by the assistance of O@CNF/NS due to the optimization of the interior crosslinking structure, as supported by FTIR, XRD, and TG results.

### 3.6. Surface Hydrophilic Property of SPI-based Films

In general, the surface of the SPI-based film should hold a high surface hydrophobicity to resist the water erosion. Hence, the surface hydrophilic property of the SPI films was investigated using a water contact angle (WCA) test, and the results are presented in Figure 7. It is found that the pristine SPI presents a small contact angle of about 51.30° due to the existence of hydrophilic groups, corresponding to a relatively high hydrophilic surface. Compared to the pristine SPI film, the introduction of CNF and CNF/NS increases the WCA to 76.45° and 58.13°, respectively, which is probably because the nanofillers form multiple physical bonding thus decreasing the amount of surface hydrophilic groups. However, the incorporated NS decreases the WCA of pristine SPI from 51.30° to 47.55° in SPI/NS, mainly due to the uneven dispersion of NS in the SPI matrix and weak hydrogen bond between the NS and SPI [34]. Most importantly, the WCA of SPI/O@CNF/NS film increases from 51.30° to 91.75° with an increment of 77.94% compared to the pristine SPI, indicating an obvious improvement of the surface water resistance even for a little water hydrophobicity. The improvement of the surface water resistance is because the KH560 not only induces CNF/NS to construct a stable crosslinking structure but also forms lots of Si–O–Si units thus endowing the protein with lower surface free energy.

## 4. Conclusions

In this study, we describe a facile nanohybrid strategy to effectively improve the overall performances of SPI-based films. The incorporated silane agent (KH560) serves as an efficient reactive platform, not only bridging CNF and NS for an excellent nanohybrid skeleton but also introducing epoxy functional groups to enhance the interfacial interactions between nanofillers and SPI matrix, which results in the formation of a stable crosslinking protein structure as confirmed by the FTIR, XRD, and SEM measurements. Compared to the pristine SPI film, the incorporation of O@CNF/NS nanohybrid increases the tensile strength value form 3.49 MPa to 6.65 MPa, realizing a 90.54% increment in the SPI/O@CNF/NS film. Meanwhile, the as-prepared SPI/O@CNF/NS film exhibits a significantly improved water resistance as compared with pristine SPI, supported by the TSM, WU, WVP, and WCA results. We envisage that the SPI/O@CNF/NS film prepared in this work can present a potential to replace non-renewable films in the development of green and renewable packaging applications.

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
