# Peer review of "Preparation and Characterization of Soy Protein Isolate-Based Nanocomposite Films with Cellulose Nanofibers and Nano-Silica via Silane Grafting"

_polymers, 2019, doi:10.3390/polym11111835_

Round 1

Reviewer 1 Report

Soy protein isolate-based nanocomposite films with a promising potential in the area of sustainable packaging technology have been prepared and studied. These nanocomposites contain cellulose nanofibers and/or silica nanoparticles. A detailed study of these films using a variety of complementary analytical techniques shows that by synergistic relationships, these films exhibit enhanced tensile strength and water resistance. Therefore, these films have significant potential in sustainable and renewable packaging applications and they could replace current non-renewable packaging films.

This manuscript is well-written and readable. The manuscript may be suitable for publication in Polymers once the following points are addressed:
1. Some characters are distorted in the pdf file, several of them in lines 155 and 156.
2. Line 192: maybe it is better to replace “curve” by “spectrum”.
3. In Fig. 2, the peak of the SPI at 19.8º is slightly shifted to higher angles when CNF and NS are incorporated. The changes observed in the XRD patterns have been attributed to the “unbalance of the ordered array of SPI molecule for a decreased crystalline degree”. CNF and NS exhibit intense peaks at around 25 and 22 º, respectively. So, is it possible that a simple superposition of the diffraction peaks of SPI, NS and CNF could occur?.
4. Line 242: “… in concordance with previous report”. Indicate references.
5. Line 264, “… a terrible interfacial stress transfer”. Please, replace “terrible” for a more technical word.
6. Authors evoke that the studied films display a technological potential as green and renewable packaging; however, these films contain nanoparticles and nanofibers that could present nanotoxicity activities. How authors demonstrate that their nanocomposites are not harmful to the environment and human health?. How the authors show that their nanocomposites are renewable?.

Author Response

Thank you for your comments. Please view the attachment for detailed modification.

Reviewer 2 Report

The manuscript entitled, "Preparation and Characterization of Soy Protein Isolate-Based Nanocomposite Films with Cellulose Nanofibers and Nano-Silica via Silane Grafting" describes the reinforcement of mechanical properties of biodegradable polymer by incorporation of nano-fibers and nano-particles in packaging technology.  The manuscript has been nicely prepared but most of the result and discussion parts are vague and speculative explanation. Herein, I would like to request the authors to revise heavily the manuscript to improve the quality of the manuscript. First of all, this work is based on the preparation of nano-composite films, but authors' have not used enough characterization technique that would give clear idea about the composite film. Although, they provide FT-IR & XRD spectra for the characterization but both the spectra and explanations are vague. In figure 1; all five spectra looks similar and authors' mentioned peak position of 1240, 1032, 914 & 910 cm-1 which are rather invisible. I would like to suggest authors to expand IR spectra, reassign all the spectra and include all other IR data for carbon nano fiber , nano silica particle.  Authors' represented the pristine SPI protein  and nano composite film TGA, and they mentioned the final composite film SPI/O@CNF/NS showed an improved thermal stability, but from figure 3 (a) & (b), the reduced degradation rate was observed for the SPI/CNF instead of final composite film, and it showed an overlap near to 400 oC, even the Ti2 and Tmax2 for final composite film  SPI/O@CNF/NS are less than the pristine SPI film. In the figure 4; authors mentioned scanning electron microscope photograph, but it is almost difficult to differentiate the effect of addition of nano hybrid particles. I would like to request authors to include EDX/EDS elemental mapping for different sets of film, that would be helpful for evaluation of nano hybrid addition. The authors mentioned the mechanical properties as a table, I would recommend authors to include Stress-strain curves as graph accompanying with table 3. In case of SPI/CNF film shows ambiguous behavior than the other, authors have not given any reasonable explanation. I would like to request them to include plausible explanation of this phenomena that would improve manuscript quality.     Throughout the manuscript, authors have mentioned several times that there are crosslink (physical and chemical) occurred into the composite film matrix, but they have not given any proof of this, and it seems like a speculation. I would suggest authors to include some experimental evidence against this query, that would up-left manuscript quality. It is well known that, the wet ability of any material signifies, more hydrophilic material absorb more water and make contact angle smaller. From figure 5, 6 and table 4 data, it was found that total soluble material, water uptake and contact angle data shows ambiguous trend. Authors have not explain the behavior, why these kind of ambiguous trend have shown by the sets of composite films.

Author Response

(The authors gave the same response as above.)

Round 2

Reviewer 1 Report

I recommend the publication of this article in the present form

Reviewer 2 Report

The authors have extensively evised the manucript, and I feel that the revised manuscript is now aceptable.